# Enzymatic Investigation of *Spongospora subterranea* Zoospore Attachment to Roots of Potato Cultivars Resistant or Susceptible to Powdery Scab Disease

**DOI:** 10.3390/proteomes11010007

**Published:** 2023-02-09

**Authors:** Xian Yu, Richard Wilson, Alieta Eyles, Sadegh Balotf, Robert Stephen Tegg, Calum Rae Wilson

**Affiliations:** 1New Town Research Laboratories, Tasmanian Institute of Agriculture, University of Tasmania, New Town, TAS 7008, Australia; 2Central Science Laboratory, University of Tasmania, Hobart, TAS 7005, Australia; 3Tasmanian Institute of Agriculture, University of Tasmania, Hobart, TAS 7005, Australia; 4Centre for Crop Health, University of Southern Queensland, Toowoomba, QLD 4350, Australia

**Keywords:** *Spongospora subterranea*, *Solanum tuberosum*, trypsin shaving, host resistance, cell-wall modification, proteome

## Abstract

For potato crops, host resistance is currently the most effective and sustainable tool to manage diseases caused by the plasmodiophorid *Spongospora subterranea*. Arguably, zoospore root attachment is the most critical phase of infection; however, the underlying mechanisms remain unknown. This study investigated the potential role of root-surface cell-wall polysaccharides and proteins in cultivars resistant/susceptible to zoospore attachment. We first compared the effects of enzymatic removal of root cell-wall proteins, *N*-linked glycans and polysaccharides on *S. subterranea* attachment. Subsequent analysis of peptides released by trypsin shaving (TS) of root segments identified 262 proteins that were differentially abundant between cultivars. These were enriched in root-surface-derived peptides but also included intracellular proteins, e.g., proteins associated with glutathione metabolism and lignin biosynthesis, which were more abundant in the resistant cultivar. Comparison with whole-root proteomic analysis of the same cultivars identified 226 proteins specific to the TS dataset, of which 188 were significantly different. Among these, the pathogen-defence-related cell-wall protein stem 28 kDa glycoprotein and two major latex proteins were significantly less abundant in the resistant cultivar. A further major latex protein was reduced in the resistant cultivar in both the TS and whole-root datasets. In contrast, three glutathione *S*-transferase proteins were more abundant in the resistant cultivar (TS-specific), while the protein glucan endo-1,3-beta-glucosidase was increased in both datasets. These results imply a particular role for major latex proteins and glucan endo-1,3-beta-glucosidase in regulating zoospore binding to potato roots and susceptibility to *S. subterranea*.

## 1. Introduction

The plasmodiophorid biotrophic pathogen *Spongospora subterranea* f. sp. *subterranea* is a significant threat to sustainable potato production wherever potato crops are grown [1]. This soil-borne pathogen infects potato tubers, underground stolons and roots, leading to tuber and root diseases [1,2,3]. Tubers infected by *S. subterranea* develop powdery scabs that affect tuber quality and storage longevity, whilst root infection affects root function (absorption of water and nutrients) and can reduce tuber yields [3]. Strategies to manage *S. subterranea* diseases are very limited. In some cases, farmers may be able to select cultivars that are relatively resistant to *S. subterranea* based on market demands; nevertheless, no cultivar is immune to infection, and substantial disease can still result in varieties that are only moderately resistant. To date, host resistance to *Spongospora* diseases has been assessed in traditional glasshouse and field trials and, more recently, using a rapid and robust in vitro zoospore root-attachment bioassay [4]. 

Infection of plant hosts by zoospores is preceded by a distinct sequence of initial zoospore recognition and attachment. Pathogen reactions to a host can be modelled on this pattern, making it a promising target for disease prevention [5,6,7]. Following attachment to a host root, *S. subterranea* zoospores discharge their contents into the plant cell walls via a particular ‘Rohr’ and ‘Stachel’ structure [1,8]. Zoosporangia form 4 to 5 d after zoospore root attachment occurs [9,10]. Our previous study showed that the efficiency of zoospore root attachment differs among potato cultivars [4]. However, the mechanisms underlying the differences in the efficiency of zoospore root attachment remain unknown. 

Previous studies on other pathosystems suggest that the molecular interactions between host-plant cell-wall surface components and the infective units of pathogens are critical in the management of pathogenesis and plant resistance [11,12]. The initiation of zoospore root attachment has been associated with the production of a range of high- or low-molecular-weight root exudates [7,13] including fucosyl residues [14,15], pectin [16,17], lectins [18], certain monoclonal antibodies [19], amino acids [20] and ions (sodium, strontium and calcium ions) [21]. Zoospore attachment to host roots by *Pythium* spp. was found to be affected by different plant polysaccharides, whereas *Phytophthora* spp. zoospore root attachment varied with the presence of pectin, polyuronates and some inorganic cations [16,21,22,23,24].

Enzymatic studies have been extensively used to examine zoospore–host interactions [11,14,15,16,25,26,27,28,29,30]. Longman and Callow [15] investigated the role of protein- and polysaccharide-based surface components involved in the attachment of zoospores of *P. aphanidermatum* to the root surfaces of cress (*Lepidium sativum*). They found that trypsin was effective in reducing the number of zoospore root attachments, as was root-surface mucus–polysaccharide modification with lectin and pectinase. Downer, Menge and Pond [29] showed that treatment with cellulase significantly reduced zoosporangia development by *P. cinnamomi* in avocado roots. However, no study has yet characterized the biochemical basis of the interaction between plant roots and *S. subterranea* zoospore attachment.

Our previous research investigated the basis of host resistance to zoospore root attachment by analysing the whole-root proteins of resistant and susceptible cultivars using label-free proteomics [31] and differential mRNA expression analysis [32]. In this study, we sought to investigate the role of protein- and polysaccharide-based root-surface components through the modification of potato roots from resistant and susceptible cultivars, using three selected enzymes (trypsin, PNGase F and cellulase). In addition, we compared the proteins identified by TS with our published whole-root proteomic analysis [31] and transcriptomic dataset for the same two cultivars [32]. A comprehensive understanding of protein profiles following TS treatment of potato roots may uncover novel targets for zoospore root-attachment control strategies.

## 2. Materials and Methods

### 2.1. S. subterranea sporosori Collection and Germination

*S. subterranea* sporosori samples were collected from powdery-scab-infected tubers of the potato cultivar ‘Kennebec’ from a commercial potato field in North-West Tasmania, Australia, 2020. Infected tubers were washed with tap water and left to air-dry in a cool and dark place for 1 to 2 d. The lesions from infected tubers were excised with a scalpel and then sifted through a 600 µm sieve. *S. subterranea* inoculum was stored at room temperature in the dark until use.

Zoospores were released by incubation of sporosori samples in Hoagland’s solution, which contained the following components: KNO_3_, 253 mg/L; Ca(NO_3_)_2_·4H_2_O, 722 mg/L; KH_2_PO_4_, 2.3 mg/L; MgSO_4_·7H_2_O, 120 mg/L; NH_4_NO_3_, 40 mg/L; Fe-EDTA, 20 mg/L; H_3_BO_3_, 140 µg/L; KCl, 400 µg/L; MnSO_4_·H_2_O, 63 µg/L; ZnSO_4_·7H_2_O, 115 µg/L; CuSO_4_·5H_2_O, 50 µg/L; and Na_2_MoO_4_·2H_2_O, 22 µg/L in deionized distilled water (DDW) [33]. Aliquots of 3 mg of sporosori inoculum were divided into 1.6 mL Eppendorf tubes and suspended in 1.0 mL of Hoagland’s solution. All tubes were incubated at 15 °C in darkness in a test chamber (Plant growth chamber, Steridium Pty Ltd., Brisbane, QLD, Australia). Zoospore release was examined by observation of subsamples (three 1 µL of subsample were examined each time, with five replicates included) by light microscopy at 200× magnification (DM 2500 LED, Leica Microsystem, Wetzlar, Germany) after 3 d of incubation [4]. 

### 2.2. Plant Materials and Growth Conditions

Tissue-cultured plantlets of the cultivars ‘Iwa’ and ‘Gladiator’ were further propagated in tissue culture in potato multiplication medium containing the following ingredients: 4.43 g/L of Murashige and Skoog (MS) salts, 30 g/L of sucrose, 0.5 g/L of casein hydrolysate, 0.04 g/L of ascorbic acid, 2.2 g/L of phytagel (pH 5.8) under a 16 h photoperiod using white fluorescent lamps (65 µmol/m^2^/s) at 22 °C. After one month, all plantlets were transferred into potato multiplication medium without the phytagel and grown for a further two weeks under a 16 h photoperiod using white fluorescent lamps (65 µmol/m^2^/s) at 22 °C.

### 2.3. Enzyme Treatments, Including the Trypsin Shaving Time-Course Study

Potato roots were collected from propagated plantlets and rinsed thoroughly with DDW. For each enzymatic treatment, six primary roots from each individual plant of each cultivar were collected from propagated plantlets and rinsed thoroughly with DDW. This experiment was performed with three technical and three biological replicates. A segment of the lower part of the root-maturation region trimmed to a length of 20 mm was selected from each individual root [4]. Three plantlets of each cultivar were used as biological replicates, thus providing a total of 18 root segments. The eighteen root segments were divided into two groups evenly (i.e., groups 1 and 2). In each group, the root segments comprising each biological replicate were added to one of three 1.5 mL Eppendorf tubes. 

A vial of 20 µg proteomic-grade trypsin (T6567; Sigma-Aldrich Pty Ltd., Macquarie Park, NSW, Australia) was dissolved in 100 µL of 50 mM ammonium bicarbonate buffer (pH 7.8) to achieve a concentration of 0.2 mg/mL. A vial of 50 units of proteomic-grade PNGase F (P7367; Sigma-Aldrich) was dissolved in 100 µL of high-purity water to provide a concentration of 500 units/mL. A quantity of 1 mg of the cellulase solution was prepared (Cellulase Onozuka™ RS, Yakult Pharmaceutical Industry Co., Ltd., Tokyo, Japan) in 1 mL of 50 mM sodium acetate buffer (pH 5.0).

For the PNGase F treatment, 45 µL of 50 mM ammonium bicarbonate buffer and 5 µL PNGase F solution (final concentration of 50 units/mL) were added to each tube in group 1. Further, all three tubes were incubated at 37 °C for 1 h [34]. 

For the cellulase treatment, 45 µL of 50 mM sodium acetate buffer and 5 µL of 1 mg/mL cellulase solution were added to each tube in group 1. Then, all tubes were incubated at 37 °C for 0.5 h [35]. 

For the TS treatment, 45 µL of 50 mM ammonium bicarbonate buffer and 5 µL 0.2 mg/mL trypsin solution (final concentrations of 20 µg/mL) were added to each tube in group 1, with 5 min of incubation at 37 °C (Sigma-Aldrich Pty Ltd., Macquarie Park, NSW, Australia). Then, the TS experiment was repeated with 15, 30, and 60 min incubations at 37 °C. 

After enzymatic treatment, all the processed root segments were assessed for in vitro zoospore root attachment using the method described below. Similarly, all root segments in group 2 (control) were assessed via in vitro zoospore root-attachment assays, directly without any pre-treatment. 

### 2.4. Spongospora subterranea Zoospore Root-Attachment Assay

All root segments were assessed according to the in vitro zoospore root-attachment assay, as previously described [4]. Root segments were placed in a plastic container (70 mm in diameter), with each replicate separated by a 100 µ mesh in the container, and then incubated in the dark at 15 °C for 48 h before further examination. The number of zoospores attached to each root segment was quantified from five randomly selected fields of view via light microscopy at 400× magnification. A preliminary study tested the effects of root-segment incubation in enzyme buffers (ammonium bicarbonate and sodium acetate) and temperature (37 °C) on zoospore root attachment, and the results showed that neither buffer nor temperature affected zoospore root attachment (data not presented). 

The zoospore root-attachment score for each cultivar/line in the screenings was normalized against the reference cultivars, ‘Gladiator’ and ‘Iwa’, with the first batch screening serving as a reference point (G1 + I1) to adjust for across-batch differences. The cultivar/line scores were further linearly scaled according to the reference-point correction coefficient (ɳ_n_) for each batch [4].
(1)$${ɳ}_{n}=\frac{\mathrm{Gn}\;+\;\mathrm{In}\;}{G1+I1}$$

### 2.5. Proteomic Analysis and Data Processing

Following TS treatment, root samples for all incubation times (i.e., 5, 15, 30 and 60 min) were prepared for proteomic analysis using C18 ZipTips (ZTC18S096; Merck Pty, Ltd., Bayswater, VIC, Australia), according to the manufacturer’s instructions. The samples were dehydrated through vacuum concentration and reconstituted in 12 µL HPLC loading buffer (2% acetonitrile and 0.05% trifluoroacetic acid in water). Thermo Scientific’s Ultimate 3000 nano RSLC system and Q-Exactive HF mass spectrometer, equipped with nanospray Flex ion source, were used to analyze peptides with nanoflow HPLC-MS/MS and Xcalibur software (ver 4.3). Three ml aliquots of each sample were initially pre-concentrated in an analytical 20 mm × 75 µm PepMap 100 C18 trapping column, followed by separation over a 60 m segmented gradient in a 250 mm × 75 µm PepMap 100 C18 analytical column kept at 45 °C, at a flow rate of 300 nL/m. The MS Tune software (version 2.9) parameters used for data acquisition were: 2.0 kV spray voltage, S-lens RF level of 60 and heated capillary set to 250 °C. MS1 spectra (390–1500 m/z) were acquired at a scan resolution of 60,000, followed by MS2 scans using a Top15 DDA method, with 20 s dynamic exclusion of fragmented peptides. MS2 spectra were acquired at a resolution of 15,000 using an AGC target of 2e5, a maximum IT of 28 ms and a normalized collision energy of 27.

Mass spectrometry raw files were processed using MaxQuant software (version 1.6.5.0), using the Andromeda search engine to search MS/MS spectra against the *Solanum tuberosum* UniProt reference proteome (UP000011115) comprising 53,106 entries. With the exception of the activation of the match-between-runs function, default parameters for mass error tolerances, missed trypsin cleavages, and fixed and variable modifications were used. The false-discovery rate was set to 0.01 for both peptide–spectrum matches and protein identifications. Protein intensity values were imported into Perseus software (version 1.6.15.0) for further analysis. Protein groups identified as potential contaminants and proteins only identified by site or by reverse database matching were removed, and LFQ intensity values were log_2_-transformed. The proteins were filtered to include only those detected in a minimum of eight samples, and remaining missing values were replaced with random intensity values for low-abundance proteins based on a normal distribution of protein abundances, using default Perseus parameters.

### 2.6. Bioinformatics and Statistical Analysis

Differentially abundant proteins were identified using a *t*-test comparison of all replicates (*n* = 12) of both resistant and susceptible cultivars, with a false-discovery rate (FDR) of 0.05 and an s0 value of 0.1 used to define significant proteins. The differentially abundant proteins were classified using the UniProt database (www.uniprot.org (accessed on 6 June 2020)), DAVID bioinformatics resources 6.8 (https://david.ncifcrf.gov/ (accessed on 1 March 2021)) and the KEGG database (www.genome.jp/kegg/ (accessed on 1 March 2021)).

Following normality and homogeneity of variance checks, all data were subjected to analysis of variance (ANOVA) using IBM SPSS Statistics 27. Zoospore root-attachment scores were analyzed using one-way ANOVA followed by protected Fisher’s LSD test to determine statistically significant differences at the 5% level (*p* = 0.05). The TS time-course incubation study revealed that zoospore root attachment was comparable for all TS incubation times (Appendix A). Therefore, only data for the 5 min incubation times are presented in the results. 

## 3. Results

In this study, we first assessed the effect of three different enzymatic treatments (trypsin, PNGase F and cellulase) on zoospore attachment to potato roots, followed by a detailed proteomic analysis of the products of the trypsin shaving treatment (Appendix A).

### 3.1. Effects of Enzyme Treatments on Zoospore Root Attachment

Zoospore root attachment was significantly reduced in root segments treated with trypsin and PNGase F when compared with the untreated control in both susceptible (‘Iwa’) and resistant (‘Gladiator’) cultivars (Figure 1). In contrast, zoospore root attachment was unaffected by cellulase for both resistant and susceptible cultivars within the enzyme concentration ranges tested. In ‘Iwa’, trypsin was the most effective treatment with respect to reducing zoospore root attachment, whilst trypsin and PNGase F both significantly reduced zoospore root attachment in ‘Gladiator’. 

### 3.2. Analysis of Proteins Released by Trypsin Shaving Treatment of Potato Roots

The ability of PNGase F and trypsin to significantly reduce zoospore root attachment highlights a potential role for proteins—in particular, *N*-linked glycoproteins—in plant–pathogen interactions. To gain a better understanding of potential mediators, we used a TS approach, in which peptides were collected from ‘Iwa’ and ‘Gladiator’ roots after incubation in trypsin for 5, 15, 30 and 60 min to allow for the detection of proteins with different susceptibilities to trypsin digestion under non-denaturing conditions. Following mass spectrometry analysis of the TS samples, a total of 1235 proteins were identified, of which 979 were quantified across the 24 samples after filtering the data to exclude proteins detected in fewer than 8 samples (Appendix A). Principal component analysis of this dataset showed that ‘Iwa’ and ‘Gladiator’ samples were separated according to PC1; however, the samples did not cluster according to time points (Figure 2a). On this basis, *t*-test analysis was used to identify differentially abundant proteins (DAPs) between the two cultivars. This analysis identified 262 DAPs, of which 132 and 130 proteins were found to be significantly higher or lower in abundance in ‘Gladiator’ compared to ‘Iwa’, respectively (Appendix A). Cluster analysis of the subset of DAPs (Figure 2b) also showed that samples collected at each time point did not cluster together, indicating that incubation time did not affect the profiles of peptides released in the TS experiment. 

### 3.3. Overall Functional Classification and Pathway Analysis of Differentially Abundant Proteins

The functional enrichment analysis of the differentially abundant proteins (resistant vs. susceptible) is shown in Figure 3a. For the proteins that were more abundant in the resistant cultivar ‘Gladiator’, the most highly enriched functional categories included glutathione transferase activity (GO_MF: 0004364), the glutathione metabolic process (GO_BP: 0006749) and the lignin biosynthetic process (GO_BP: 0009809). For the proteins that were reduced in the resistant cultivar, significant functional categories included protein heterodimerization activity (GO_MF: 0046982), ATPase activity (GO_MF: 0016887) and nucleosome assembly (GO_BP: 0006334). 

Pathway analysis revealed alterations in metabolic pathways in both subsets of DAPs (Figure 3b). Specific pathways associated with the proteins that were increased in the resistant cultivar included oxidative phosphorylation (*n* = 8 proteins), biosynthesis of nucleotide sugars (*n* = 7 proteins), and amino sugar and nucleotide sugar metabolism (*n* = 7 proteins). In contrast, proteins that were less abundant in the resistant cultivar were related to carbon metabolism (*n* = 12 proteins), carbon fixation in photosynthetic organisms (*n* = 8 proteins), glyoxylate and dicarboxylate metabolism (*n* = 6 proteins), and the pentose phosphate pathway (*n* = 5 proteins). 

### 3.4. Comparison of Proteins Identified by TS with Whole-Root Proteomics and Transcriptomics

The bioinformatic analysis of the complete set of DAPs identified by the TS experiment identified significant functional differences between the proteomes of resistant and susceptible cultivars. However, this included a high proportion of cellular components that may not be directly involved in facilitating attachment to root surfaces. Therefore, we used our previous proteomic dataset acquired from whole-root tissue analysis to filter the TS dataset [31], which enabled us to identify a subset of 226 proteins that were unique to the TS experiment (Figure 4a). Interestingly, a high proportion of these proteins (188) were significantly different in terms of abundance between resistant and susceptible cultivars (Figure 4b).

Of the 188 significant proteins that were unique to the TS dataset, 92 were more abundant in the resistant cultivar, while 96 were less abundant. Proteins that were detected at increased levels included globulin (M1C704), ER6 protein (M1AZC6) and B12D protein (M0ZLR3), while those that were reduced included wound-induced proteinase inhibitor 1 (P08454) and major latex proteins (M1CYU9 and M1BBE7) (Figure 4c). Of note, the cell-wall stem 28 kDa glycoprotein was significantly less abundant in the resistant cultivar, whilst three glutathione *S*-transferases (GSTs) (M0ZQ26, M1ARE1 and M0ZQ38) were significantly more abundant in the resistant cultivar specific to the TS experiment (Figure 4c). Fifty-nine proteins specific to the TS dataset were also altered in abundance due to differential expression at the mRNA level, based on a comparison with our previously published transcriptomic analysis of the cultivars Iwa and Gladiator [32]. The relative differences (log_2_FC, resistant vs. susceptible cultivars) in their transcript and protein levels were compared (Figure 4d). Thirty-nine proteins underwent changes in abundance that were in agreement between the two datasets, while 20 proteins underwent opposite changes in abundance between the RNA-seq and proteomic data. Globulin (M1C704) and two glutathione *S*-transferases (GSTs) (M0ZQ26 and M0ZQ38) were among the proteins that were found at increased levels in both datasets.

Further comparison of the TS dataset with the whole-root proteome analysis enabled us to identify proteins with consistently large changes in abundance in both datasets. We selected the 20 proteins with the greatest differences in abundance in the TS treatment (ten increased and ten reduced in resistant vs. susceptible cultivars), of which 17 were also identified in the whole-root proteomic dataset. The fold changes (log_2_) for these proteins are compared in Figure 5, where the values for the TS dataset are plotted against the respective values for the whole-root proteomic dataset (Appendix A show all proteins and significant DAPs, respectively). The protein with the largest increase in the resistant cultivar (Glucan endo-1,3-beta-glucosidase: P52401) was highly modulated in both datasets (4.4-fold in the TS data and 6.0-fold in the whole-root proteomic data). Globulin (M1C704) was also significantly increased in the resistant cultivar in both datasets. Conversely, the Wound-induced proteinase inhibitor 1 (P08454) showed the largest decrease in abundance, with 5.3-fold and 3.7-fold reductions in the TS treatment and whole-root samples, respectively. Major latex protein (M1AFT2) and an uncharacterized protein (M1AXR4) were also consistently and significantly decreased in the resistant cultivar. Notably, only one protein, abscisic acid- and environmental-stress-inducible protein (M0ZVK4), showed opposite trends in the TS and whole-root proteomic datasets, with a 3.3-fold decrease and a 3.7-fold increase, respectively.

## 4. Discussion

In this study, a combination of the in vitro zoospore root-attachment assay and label-free proteomic analysis was used to investigate pathogen–host interactions based on the modification of plant root-surface components with specific enzymes. We showed that trypsin and PNGase F, assessed in an in vitro model in this study, both reduced *S. subterranea* zoospore attachment to potato roots. PNGase F is an enzyme that catalyzes the removal of *N*-linked oligosaccharide chains from glycoproteins in a full and efficient manner. This enzyme is commonly used to investigate structure–function relationships of glycoproteins [36]. Plant cell-wall polysaccharides and proteins may serve as inactive signal molecules during plant–pathogen interactions [37,38]. Several studies have reported the biochemical basis of zoospore root attachments and demonstrated that root-surface polysaccharides play a critical role in zoospore root recognition and attachment [13,15,18,19,23,27,39,40]. The effects of plant cell-wall proteins and polysaccharides on *Pythium* and *Phytophthora* zoospore host attachment have been demonstrated previously [6,12]. The removal of polysaccharides of cress (*Lepidium sativum*) from root surfaces resulted in a reduction in *Pythiaceous* zoospore attachment; treatments that block or remove terminal fucosyl residues were particularly effective [15]. Similarly, Estradagarcia et al. [13] confirmed that cress-root mucilage can encourage the process of zoospore root attachment. In the present study, while cellulase had no effects on inhibiting zoospore root attachment, both trypsin and PNGase F significantly decreased the attachment of *S. subterranea* zoospores to the roots of two potato cultivars (Figure 1). These results suggests that potato root proteins, especially *N*-glycoproteins, may impact the zoospore root-attachment process. 

Following the preliminary assessment of the effect of enzymatic treatment on root attachment, which indicated the potential involvement of cell-surface proteins, we used the trypsin shaving approach as the most practical first step towards the identification of cultivar-specific glycoproteins. Among the 1235 proteins identified in the TS study, most of the proteins that were significantly increased in the resistant cultivar were associated with metabolic pathways, such as oxidative phosphorylation, biosynthesis of nucleotide sugars and the majority of amino acid biosynthesis pathways (Figure 3). These proteomic results were in line with the findings of similar proteome analyses of rice and sweet potato [41,42]. According to the analysis of pathways and GO functional annotation, we observed that glutathione metabolism, including the glutathione metabolic process and glutathione transferase activity, occurred at a high rate in the resistant cultivar compared to the susceptible cultivar. Glutathione biosynthesis occurs in chloroplasts, cytosol and mitochondria [43,44]. A few studies revealed the critical role of glutathione-related enzymes in host resistance to different pathogen infections. For example, glutathione-related enzymes were abundant in a tomato cultivar resistant to *Oidium neolycopersici* [45] and a rapeseed cultivar resistant to *Sclertinia sclerotiorum* [46]. Three GST proteins were found to be highly abundant in the resistant cultivar specific to the TS study, while two of them were also more abundant according to the resistant cultivar’s RNA-seq data. Balotf et al. [32] reported that GST proteins were significantly abundant in the roots of a resistant potato cultivar after *S. subterranea* infection. In the potato genome, there are at least 90 GST proteins that are involved in the plant immune system [47]. In a study of the interaction between *S. subterranea* and potato, it was shown that more than 30 GST genes were induced after infection [32]. 

In our present study, we compared the proteomes of root cell surfaces of two potato cultivars in the absence of *S. subterranea* infection and concluded that both constitutive and responsive gene expression strategies are involved in the regulation of GST proteins and used by potato hosts to increase resistance to *S. subterranea*. Lignin biosynthesis processes were also identified in our functional analysis of DAPs in the resistant cultivar. Lignin serves as a crucial barrier against pest and pathogen infection [48]. In our previous study [4], the phenylpropanoid biosynthesis pathway was identified in resistant cultivars associated with *S. subterranea* zoospore root attachment. Similar results were obtained by Balotf et al. [32], in whose study the phenylpropanoid metabolic pathway and especially lignin biosynthesis were shown to play important roles in the constitutive resistance of potato to *S. subterranea*. 

The in vitro zoospore root-attachment assay (Appendix A) indicated that a 5 min incubation was sufficient for the enzyme to take effect, while, with respect to the time course for TS, no significant differences were found between incubation times (Appendix A). Elsewhere, He and De Buck [49] reported that digestion of cell-wall proteins of *Mycobacterium avium* subsp. *paratuberculosis* with trypsin required 30 min and a temperature of 37 °C. In contrast, Zahir, et al. [50] found that intracellular proteins were detected only after increasing the trypsin incubation period from 30 min to 60 min. In our investigation, cytoplasmic proteins were detected after trypsin shaving at all time points tested, suggesting that further optimization is required to increase the specificity for cell-surface proteins. However, we were able to use our whole-root proteomic analysis to filter the TS dataset and target potential cell-surface peptides of interest.

Comparison of proteins from the TS proteome study with the whole-root proteins revealed 188 DAPs that were significantly abundant in the TS treatment (Figure 4b). Major latex proteins (M1CYU9 and M1BBE7), which play crucial roles in plant defence, were significantly reduced in abundance in the resistant cultivar. The major latex protein (M1AFT2) was also found to be consistently reduced in abundance in the resistant cultivar via both TS treatment and whole-root protein analysis. Major latex proteins exist in different plant species, such as opium poppy [51,52], cucumber [53], peach [54], melon [55], soybean [56] and grape [57]. The number of major latex proteins varies among species; for instance, *Arabidopsis thaliana* contains 24 major latex proteins, while grape has just 14 [57,58]. Major latex proteins respond to biotic and abiotic stressors and perform crucial roles in plant growth and development, such as disease resistance, stress tolerance and development [59,60]. He, et al. [61] revealed that major latex proteins negatively regulate resistance to fungal infection in apple (*Malus domestica*) by suppressing the expression of genes and transcription factors associated with defence and stress. Similar to this result, we showed that the resistant potato cultivar, ‘Gladiator’, had lower expression of major latex proteins than the susceptible cultivar ‘Iwa’.

The cell-wall stem 28 kDa glycoprotein (Figure 4c) was another protein that was found to be less abundant in the resistant cultivar, which was consistent with the whole-root protein analysis [31]. Previous studies have reported that stem 28 kDa glycoprotein plays a critical role in the transformation of immature elongation regions into mature, thickening tissues in the youngest regions [62]. Glycosylation and glycan processing are crucial post-translational modifications that cell-wall proteins undergo within the cell and are regarded as essential for the control of growth and defence mechanisms in plants [63]. PNGase F treatment demonstrated that *N*-glycoproteins can suppress the zoospore root attachment considered in this study. Together with these results, the cell-wall stem 28 kDa glycoprotein is an interesting candidate for direct association with the susceptibility of potato roots to zoospore root attachment.

Glycan endo-1,3-beta-glucosidase (P52401) was the protein with the largest fold change, being identified as significantly more abundant in the resistant cultivar with the TS treatment, and this finding is consistent with the whole-root protein analysis (Figure 5). Glucan endo-1,3-beta-glucosidase is a type of hydrolytic enzyme that breaks down 1,3-β-D-glucosidic linkages in β-1,3-glucans, which exist widely in bacteria, fungi and viruses [64]. Shinshi, et al. [65] reported that tobacco glucan endo-1,3-beta-glucosidase displays complicated hormonal and developmental regulation and is triggered by pathogen infection. In line with these studies, our results indicated that glucan endo-1,3-beta-glucosidase contributes to potato resistance against *S. subterranea* infection. The abscisic acid- and environmental-stress-inducible protein (M0ZVK4) was one of the proteins that was differentially changed between the TS and whole-root protein analysis. This protein decreased in the TS treatment but increased in the whole-root protein analysis for the resistant cultivar. Abscisic acid is essential for numerous cellular processes, including seed development, germination, crop growth and root architecture mediation [66,67]. According to Harris [67], abscisic acid mediates responses to various environmental factors, including the presence of nitrate in the soil, water stress and salt, shaping the root system by regulating the production of lateral roots and controlling root elongation by modulating cell division and elongation. Since only the lower part of the mature potato root was examined in the TS treatment, while the entire root was used in the whole-root protein analysis, the difference in the fold changes of the protein (M0ZVK4) between the two studies may be a consequence of different spatial distributions of abscisic acid across different root areas.

## 5. Conclusions

This is the first report of an investigation of the biochemical basis of potato root-surface components in relation to *S. subterranea* zoospore attachment. From the in vitro zoospore root-attachment study, the enzymes trypsin and PNGase F were found to significantly reduce zoospore root attachment, whilst cellulase had no effect on zoospore root attachment. Our detailed proteomic analysis revealed broad-scale differences of root proteins between susceptible and resistant potato cultivars. These proteins within potato roots provide new insights into host resistance to zoospore root attachment at a proteomic level. Overall, this study provides an initial understanding of the biochemical and molecular bases of potato resistance to zoospore root attachment and is important for developing novel approaches in future disease management.

This work contributes to knowledge of the biochemical and molecular bases of *S. subterranea* zoospore root attachment, but there are some limitations that ought to be mentioned. Firstly, the TS peptide analysis identified a large number of intracellular proteins, which may have hindered the identification of lower-abundance cell-surface proteins. While different time points were assessed in this study, further refinement of the TS approach may help to minimize the background of intracellular proteins. Secondly, trypsin shaving is inherently a peptide-centric approach that cannot easily distinguish between different proteoforms and therefore may underestimate proteome complexity. However, future studies using glycoproteomics may lead to a better understanding of the role of protein glycosylation in cultivar resistance to zoospore root attachment. 

## Figures and Tables

**Figure 1 proteomes-11-00007-f001:**
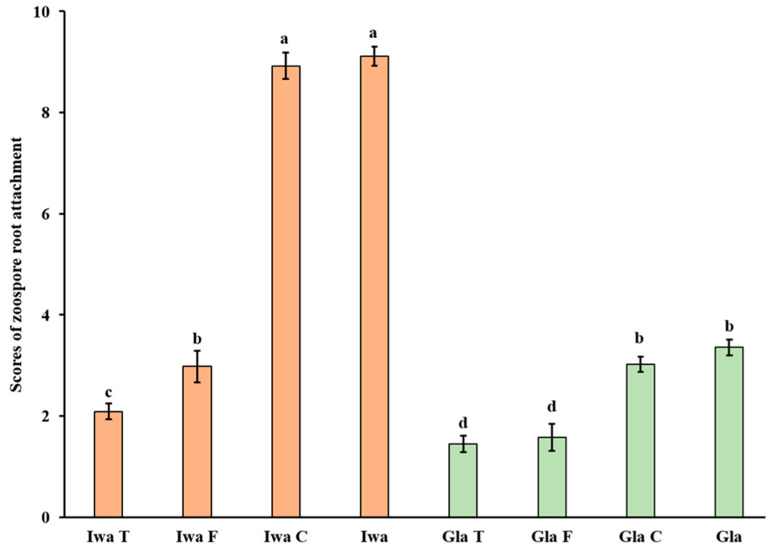
Effects of pre-enzyme treatment on zoospore root attachment for the ‘Gladiator’ (Gla, green bars) and ‘Iwa’ cultivars (orange bars). T: trypsin (20 µg/mL); F: PNGase F (50 units/mL); C: cellulase (1 mg/mL). Error bars represent standard deviations based on three biological replicates. Lower case letters denote values that are significantly different from each other. *p* (cultivars) < 0.001, *p* (treatment) < 0.001, *p* (cultivar × treatment) < 0.001. LSD (0.05) = 0.45.

**Figure 2 proteomes-11-00007-f002:**
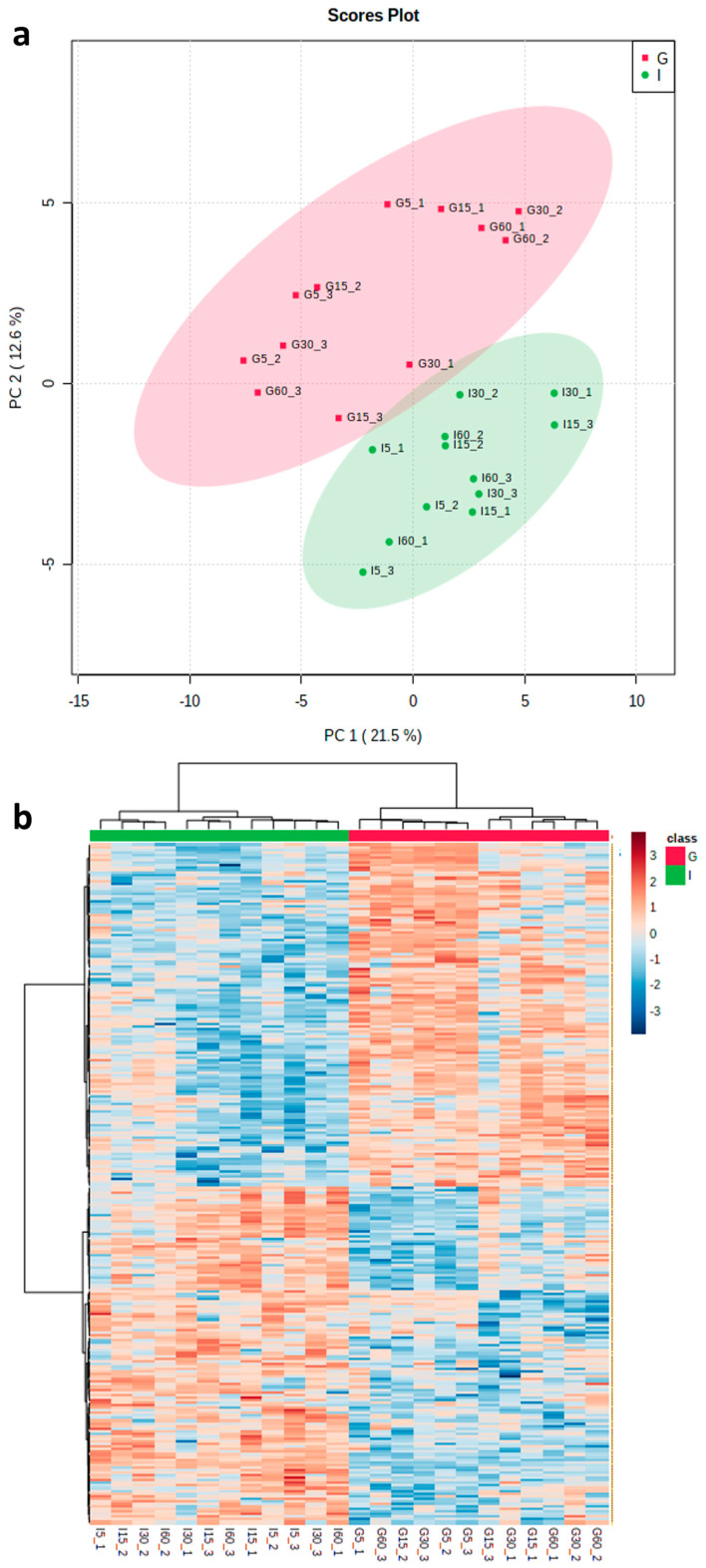
(**a**) Principal component analysis (PCA) of all identified proteins from both resistant and susceptible cultivars (*n* = 3) at four incubation times (5, 15, 30 and 60 min). (**b**) Heatmap analysis of all significantly abundant proteins (‘Gladiator’ vs. ‘Iwa’) at four incubation times (5, 15, 30 and 60 min). G: ‘Gladiator’; I: ‘Iwa’.

**Figure 3 proteomes-11-00007-f003:**
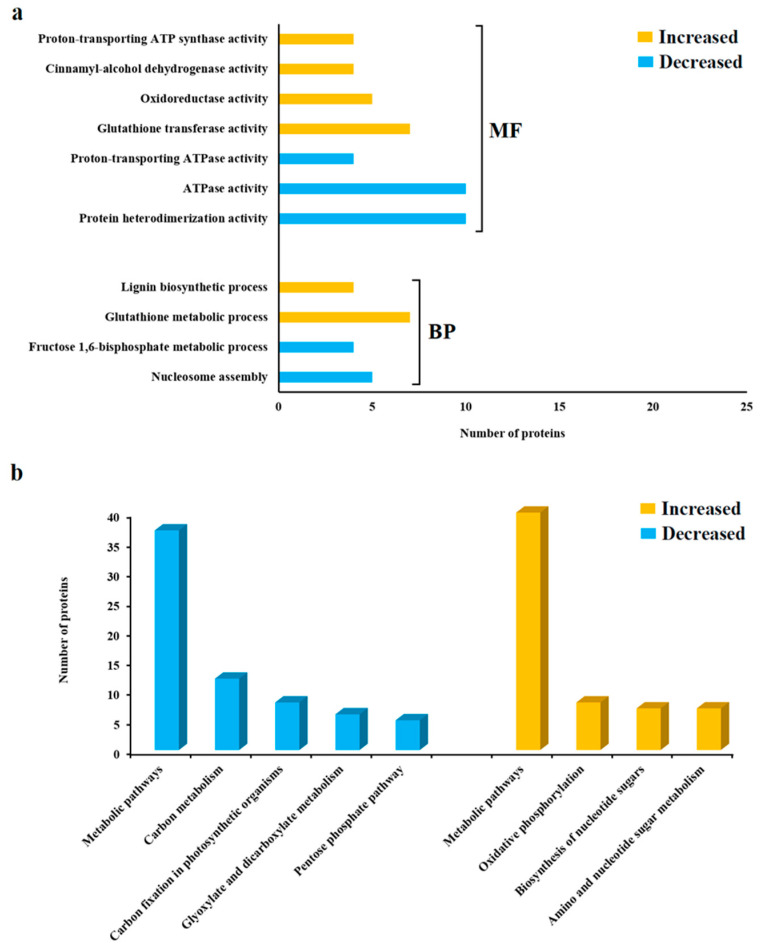
(**a**) Gene ontology (GO) categories of DAPs (‘Gladiator’ vs. ‘Iwa’) from potato roots as determined with the trypsin shaving (TS) treatment. BP: biological process; MF: molecular function. (**b**) Pathway classification and enrichment analysis of DAPs (‘Gladiator’ vs. ‘Iwa’) from potato roots via TS treatment.

**Figure 4 proteomes-11-00007-f004:**
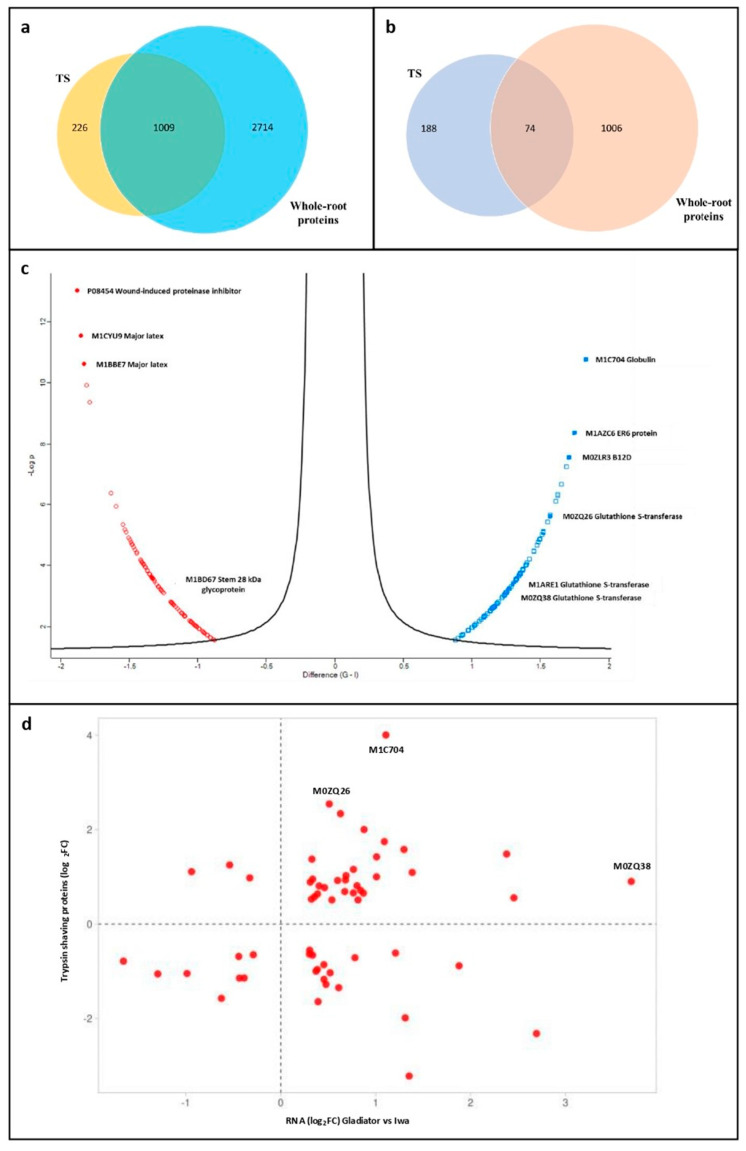
(**a**) Venn diagram representing the total number of potato root proteins identified specifically in the trypsin shaving (TS) treatment or the whole-root proteome analysis or in both (overlap). (**b**) Venn diagram representing the subsets of significant potato root proteins (resistant vs. susceptible DAPs) identified specifically in the TS treatment or whole-root proteome analysis or both (overlap). (**c**) Volcano plot displaying the 188 significant DAPs (resistant vs. susceptible) specific to the TS treatment plotted according to their log_2_ fold changes (*t*-test differences) on the x-axis and -log_10_ *p*-values on the y-axis. Data points in blue represent the proteins significantly increased in the resistant cultivar and those in red the proteins significantly increased in the susceptible cultivar. (**d**) Scatter plot representing the subset of 59 proteins that were significantly altered at both the mRNA and protein levels. Data points are displayed as the log_2_ fold changes (resistant vs. susceptible) at the mRNA level on the x-axis vs. the log_2_ fold changes (resistant vs. susceptible) at the protein level on the y-axis. The three datapoints labelled with their accession numbers are Globulin (M1C704) and two glutathione S-transferase proteins (M0ZQ26 and M0ZQ38).

**Figure 5 proteomes-11-00007-f005:**
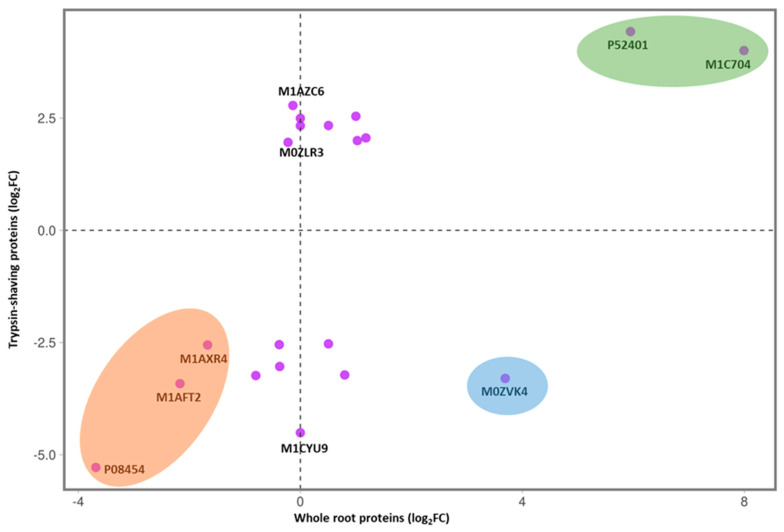
Comparison of the ten proteins with the largest increased or decreased fold changes (log_2_) in the resistant cultivar (‘Gla’) from the trypsin shaving (TS) treatment and the whole-root protein analysis. Proteins in green ellipse: most significantly increased in resistant cultivar in TS treatment and whole-root proteins; proteins in orange ellipse: significantly decreased in resistant cultivar in TS treatment and whole-root proteins; proteins in blue ellipse: significantly increased in resistant cultivars in whole-root proteins and significantly decreased in resistant cultivars in TS treatment.

## Data Availability

The mass spectrometry proteomic data have been deposited with the ProteomeXchange Consortium via the PRIDE partner repository with the dataset identifier PXD037060. The whole-root proteomic data are publicly available via ProteomeXchange with the identifier PXD022502.

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
