# Peer review of "Enzymatic Investigation of Spongospora subterranea Zoospore Attachment to Roots of Potato Cultivars Resistant or Susceptible to Powdery Scab Disease"

_proteomes, 2023, doi:10.3390/proteomes11010007_

Round 1
Reviewer 1 Report
This study analyses the proteins and polysaccharides present on the surface of the roots and their role in the zoospores attachment of S. subterranean in the infection process. For this, a susceptible and a resistant cultivar were used. Enzymes were used to remove proteins and polysaccharides from the surface and then, the amount of attached zoopores was evaluated. An analysis of the proteins present in root segments by trypsin shaving was also performed.
Results showed that cellulase had no effects on inhibiting zoospore root attachment, but trypsin and PNGase F significantly decreased the attachment of S. subterranea zoospore to the roots in susceptible and resistant potato cultivars. In addition, it was observed that proteins related with Specific pathways associated increased in the resistant cultivar (oxidative phosphorylation, biosynthesis of nucleotide sugars and amino sugar and nucleotide sugar metabolism, but proteins that were less abundant in the resistant cultivar were related to carbon metabolism, carbon fixation in photosynthetic organisms, glyoxylate and dicarboxylate metabolism and the pentose phosphate pathway. Also, it is discussed the rol of glucan endo-1,3-beta-glucosidase in defence mechanism again S. subterranea, it was one of the largest fold change proteins identified as significantly more abundant in the resistant cultivar with the TS treatment, and this finding is consistent with the whole-root protein analysis
In general, it is a novel article, very interesting, with significant content, methodology and results are explain very well. It also contains a very good and actual reviewed.
I only suggest to do a better description of the information in Figure 4. The explanation of the content is confusing.
Also, in line 225, it said. … were quantified across the 24 samples…. What are these samples? these came from the group 2?
Reviewer 2 Report
The authors report a comparative proteomic analysis between one susceptible and one resistant potato cultivar to Spongospora subterranea. This study adds interesting new information on the biochemical and molecular basis of zoospore attachment to root surfaces and points out proteins involved in this process. The manuscript is very well written and understandable. I am only pointing out a few changes that need to be made.
Decease the number of abbreviations in the text - for example: HS, TS, PM, MF, BP, etc. They make understanding difficult and do not significantly contribute to cut the length of the manuscript.
The authors did not mention among the limitations of the study a confirmation of gene expression with qPCR. Isn’t it worth mentioning?
L266 - I know what the authors mean, but the wording is strange - root attachment to the cell surface - Please, improve.
L320-321 - there is only only one resistant cultivar
L331 - dormant signals sound strange - can the authors improve it?
L338-339 - references are cited in a strange way compared to L378 and L359, which do not agree with the reference list- please check these and also the whole manuscript. I honestly do not know the format required by Proteomes.
L379-381 - please, be consistent in the way of writing min.
L419-420 - In line with ….
L422 - …one of the proteins …
Reviewer 3 Report
Yu et al., reported the underlying mechanisms of resistant/susceptible to zoospore attachment in potato.Here are some comments.
1. Please add an experimental design as supplemental figure to show the study clearly.
2.line 162, "reconstituted in 12 mL HPLC loading buffer"? How many proteins were used for the proteomics study? Here, are you sure it was 12 mL HPLC loading buffer used for the reconstituted?
3. Figure1 legend showed p(cultivars)<0.001, however, I didn't see P character in fig1. And what is the meaning of y-axis? the scores of zoospore root attachment. How did you get the score?
4.The authors use three different enzyme for the treatment(fig1). Finally, the trypsin was chosen for the further study. Please add some discussion for the trypsin selection in the discussion part. Specially for the choice of trypsin not PNGase F.
